# A Probabilistic Model of Human Activity Recognition with Loose Clothing [note 1]

**DOI:** 10.3390/s23104669

**Published:** 2023-05-11

**Authors:** Tianchen Shen, Irene Di Giulio, Matthew Howard

**Affiliations:** 1Centre for Robotics Research, Department of Engineeing, King’s College London, London WC2R 2LS, UK; 2Centre for Human and Applied Physiological Sciences, King’s College London, London SE1 1UL, UK

**Keywords:** wearable sensing, human motion detection and tracking, electronic textile

## Abstract

Human activity recognition has become an attractive research area with the development of on-body wearable sensing technology. Textiles-based sensors have recently been used for activity recognition. With the latest electronic textile technology, sensors can be incorporated into garments so that users can enjoy long-term human motion recording worn comfortably. However, recent empirical findings suggest, surprisingly, that clothing-attached sensors can actually achieve *higher* activity recognition accuracy than rigid-attached sensors, particularly when predicting from short time windows. This work presents a probabilistic model that explains improved responsiveness and accuracy with fabric sensing from the increased statistical distance between movements recorded. The accuracy of the comfortable fabric-attached sensor can be increased by 67% more than rigid-attached sensors when the window size is 0.5s. Simulated and real human motion capture experiments with several participants confirm the model’s predictions, demonstrating that this counterintuitive effect is accurately captured.

## 1. Introduction

Human motion analysis is important in many different fields of study. For instance, human–robot interaction [1], physical rehabilitation and medical care [2]. The recent development of electronic textiles (e-textiles) has made it possible to incorporate sensors into garments [3]. This has significant advantages, such as the ability to capture *natural* behavior and ensure the wearer’s comfort through unobtrusive sensing, and the sensors can be attached to any position on the clothing [4].

However, one of the issues with clothing-embedded sensing is the additional motion of the fabric’s movement with respect to the body (see Figure 1). The prevailing view is that this motion needs to be treated as an unwanted artifact that should be eliminated. For this purpose, several approaches to remove or limit it have been suggested, such as (i) ensuring a rigid attachment between sensor and body [5], (ii) supervised errors-in-variables regression [6], (iii) unsupervised latent space learning [7] (iv) and difference mapping distributions [8]. However, several recent works focus on fabric motion: (i) Michael and Howard [9] found that fabric motion may actually *assist* human motion analysis, particularly in activity recogntion (AR). This could be due to some possible factors (e.g., uncertainty in the fabric itself, movement frequency, etc.). (ii) Jayasinghe et al. [10] discovered that the movement of clothing can be useful to describe human daily activities (walking, running, sitting and riding a bus) and gait analysis [11]. However, this phenomenon so far lacks a satisfactory theoretical explanation or model.

To this end, this paper proposes a probabilistic framework as a basis for understanding this phenomenon. A probabilistic model is introduced in which it can be shown that statistical distance measures such as the Kolmogorov-Smirnov (KS) test imply that stochastic fabric movements lead to greater discriminative ability. The predictions of the model are verified in a set of simulated and real human motion capture experiments, where it is evident that sensors loosely attached to fabric yield greater accuracy than rigidly attached ones, especially when making predictions under time pressure. Despite the simplicity of the model, empirical data show it is surprisingly accurate at capturing the aforementioned phenomenon in a variety of conditions. This suggests that it could be a useful tool in the design and analysis of motion capture systems using ordinary garments that allow for enjoyment of their comfort and user acceptability.

The remainder of this paper is organised as follows. Section 2 reviews previous work in machine learning applied to fabric-based sensing. Section 3 introduces a probabilistic model to predict the effect of fabric movement on statistical AR. Section 4 evaluates the model predictions as applied to physical motion recognition tasks. Section 5 provides a summary and discussion of the findings.

## 2. Related Work

There are numerous research papers focusing on human motion analysis based on wearable sensing [12]. In the majority of these studies, sensors are rigidly mounted on the body using tape, glue, straps, etc. Data analysis is strictly focused on body movement [10]. Relatively few studies investigate the case of motion signals arising from loose-clothing-attached sensors. Table 1 compares a selection of studies that involve fabric/clothing-based movement sensing and analysis. Some studies [13,14,15,16] focus on how to embedded the sensor into clothes using different e-textile technologies. Others [9,10,11,17] apply machine learning algorithms to sensor readings. However, they have not compared the performance of AR of body-attached and clothing-attached sensors with human movement, and they lack probabilistic models to understand the performance of AR of sensors with two different attachments.

Jayasinghe et al. [10] investigated and quantified the extent to which data obtained from clothing-attached sensors can be used to characterize activity compared with body-attached sensor data. Three pairs of sensors (accelerometers) were attached to three segments of the body (waist, thigh and ankle) and clothing (slacks, pencil skirt and loose frock) in similar positions and used to record daily activities (e.g., walking, running and sitting). The results show that clothing sensor data are strongly correlated with body-attached sensor data. In particular, the signal collected from the pencil skirt attached tightly at the waist had the highest correlation, whereas that collected from the loose frock had low correlation because motion artifacts formed a greater part of the signal. However, the authors did not analyze how those artifacts might contribute toward *enhancing* AR accuracy. Moreover, the study focused on activities that are relatively easy to distinguish (walking, running and sitting). Recently, Jayasinghe et al. [11] extended this work to focus on the analysis of gait-related movements (e.g., standing, sitting, sit-to-stand, stand-to-sit, leg raises and walking back and forth) using sensorized trousers with similar findings reported. Jayasinghe et al. [17] recently further explored the practicability of clothing-attached sensors to classify static postures using K-nearest neighbors (KNN) and achieved a promising result.

Michael and Howard [9] directly compared the performance of AR between rigid- and fabric-attached sensors using an instrumented pendulum. Three types of fabric (i.e., denim, roma and jersey) were affixed to the tip of a pendulum and inertial sensors were mounted such that one measured the movement of the tip of the pendulum (i.e., measurement from a rigidly attached sensor), one measured the movement of the the middle of the fabric and a third measured movement at the edge of the fabric. Data from these were used to train support vector machine (SVM) and discriminative regression machines (DRM) classifiers to distinguish between movements with small differences in frequency. Surprisingly, the results show higher accuracy in the classifiers trained on the data from the fabric-attached sensors. This study was the first to suggest that fabric movement may be beneficial to AR but lacked a sound theoretical model as to the cause of this effect. This lack of a model makes it hard to predict the accuracy of AR under different situations, such as how and when AR may benefit from fabric movement. Moreover, these findings relied on a pendulum as the experimental platform, so they lacked verification that the effect is reproduced in more realistic settings, such as the AR of humans wearing real clothing.

This paper proposes a probabilistic model and verifies it in a physical realization and a real human AR task. The experiment result shows that this model can be used to predict the performance of AR in different situations. This study is also the first to suggest that fabric movement can be beneficial to AR in a real human task.

## 3. Probabilistic Modeling Framework

The following introduces the proposed framework and presents example of its application to a simple movement classification task.

### 3.1. Problem Definition

Activity recogntion is defined as the classification of movements into a number of discrete categories of activity (e.g., walking, running, etc.) based on motion data. For simplicity, in the following, it is assumed that the motion data, Y, consist of N measurements of the absolute position of a point $p_{y}$ on a garment, collected over an extended duration of time at a regular sampling frequency. In AR, each data point is also associated with a label (throughout the paper, without loss of generality, the class labels are assumed to be binary) c∈{0,1} corresponding to the category of movement, so $Y=\{\left(y_{1},c_{1}\right),\ldots ,\left(y_{N},c_{N}\right)\}$. This is contrasted with so-called *rigid data,*U, recorded under the same conditions, except that the sensor is rigidly attached to the moving body. The goal of AR is to train a classifier on Y such that, when presented with (previously unseen) movement data $y^{*}$, the corresponding class label $c^{*}$ can be accurately predicted.

As discussed in Section 2, previous studies have provided empirical evidence that, contrary to expectations, AR performance is *improved* when using data from *loose fitting garments* [9]. This paper proposes a probabilistic model that predicts this effect.

### 3.2. Probabilistic Model of Fabric Motion

The effect of fabric motion on the data Y is subject to a high degree of uncertainty, arising from challenges in estimating the fabric’s physical properties and the resultant movement complexity. To deal with this, it is proposed to model the data generation process through stochastic methods.

Specifically, the position y of the point $p_{y}$ on the fabric at any given time is modeled as the stochastic process consisting of the corresponding position u of a point $p_{u}$ on the rigid body plus a random offset δ introduced by the fabric motion. In the univariate case, this can be written as
(1)Y=U+Δ
where y(t)∼Y, u(t)∼U and δ(t)∼Δ.

The key to determining whether the fabric movement is beneficial to AR is to understand the effect of Δ on the statistical distance between movement classes. A condition for improved classification performance is a greater statistical distance in data distribution between movement classes in Y compared with U, i.e.,
(2)$$D\left(Y_{c=0},Y_{c=1}\right)>D\left(U_{c=0},U_{c=1}\right)$$
where $Y_{c=0}$ is the distribution of fabric data for movement class c=0, $Y_{c=1}$ is the same for movement class c=1, $U_{c=0}$ and $U_{c=1}$ are the equivalent distributions for rigid data and D(·,·) is a suitable metric in probability space. For the latter, several choices are available (e.g., Kullback-Leibler (KL) divergence or Jensen-Shannon (JS) divergence). In the following, the KS metric is used [21]
(3)$$\begin{array}{cc} \; & D\left(Y_{c=0},Y_{c=1}\right)=\sup\left|F\left(Y_{c=0}\right)-F\left(Y_{c=1}\right)\right|\\ \; & D\left(U_{c=0},U_{c=1}\right)=\sup\left|F\left(U_{c=0}\right)-F\left(U_{c=1}\right)\right| \end{array}$$
where F(·) denotes the CDF of a random variable. Note that KS is chosen since the range of possible fabric positions may differ depending on the movement, meaning that $Y_{c=0}$ and $Y_{c=1}$ occupy different probability spaces, preventing measures such as KL or JS from being computed. In the next section, a simple working example is presented to illustrate the effect of fabric motion on activity recogntion predicted by this model.

### 3.3. Example: Oscillatory Motion

Consider the problem of movement analysis for a one-dimensional, scotch yoke mechanism using data from a sensor mounted on an inextensible piece of fabric attached to the mechanism (see Figure 2a). Here, an AR task might involve classifying a set of movements of the rigid body (e.g., those with different frequencies) from raw positional data.

In this system, the horizontal position of the rigid body (point $p_{u}$) over time is given by
(4)u(t)=asin(ωt+ψ)
where *a* is the amplitude, ω is the frequency (equivalently, the angular speed of the rotary wheel of the mechanism) and ψ is the phase (equivalently, the starting angle of the wheel), and (without loss of generality) it is assumed a=1 and 0<L≤1. In this system, samples of the rigid body position follow a beta distribution
(5)$$U∼B(\frac{1}{2},\frac{1}{2}).$$

As the rigid body moves, the fabric will move alongside it but undergo additional displacement due to the complex fabric dynamics according to (1). In general, the nature of these displacements will depend on (i) its physical properties (e.g., mass distribution, stiffness, fiber structure, length, width and orientation) and (ii) the movement pattern of the rigid body (e.g., amplitude, frequency). Noting that here the fabric is *inextensible*, the maximum possible displacement δ of the point $p_{y}$ from $p_{u}$ is *L*, suggesting that (in the absence of other prior information) a simple choice of its distribution could be
(6)Δ∼(−L,L).However, this would imply that any displacement −L≤δ≤L is equally likely, whereas in practice, δ tends to be greater when there is greater excitation of the fabric by the movement of the yoke, e.g., for higher-frequency movements (see Figure 3a,b). Therefore, the following assumes
(7)Δ∼(−zL,zL),
where
(8)$$z=1-\exp(-\omega^{2}).$$Note that this respects the constraint that −L≤δ≤L (since (8) causes 0≤z≤1), while capturing the tendency for δ to increase at higher frequencies up to a saturation point. Note also that the modeling approach is easily extensible to take account of more factors according to their relevance; for instance, wind of a constant direction and speed would introduce a bias in the fabric movements, which could be included by introducing an offset to the expectation of Δ.

According to this model, the CDF of *U* is [22]
(9)$$F\left(u\right)=\left\{\begin{array}{cc} 0 & u<-1\\ \frac{\sin^{-1}\left(u\right)}{\pi }+\frac{1}{2} & -1\leq u\leq 1\\ 1 & u>1 \end{array}\right.$$
and it can be shown (a derivation is provided in Section A.1.1 and Section A.1.2) that the CDF of *Y* is dependent on zL, with two possible cases.

*Case 1* (0<zL<1):(10)$$F\left(y\right)=\left\{\begin{array}{cc} 0 & y<-1-zL\\ F_{1}\left(y\right) & -1-zL\leq y<-1+zL\\ F_{2}\left(y\right) & -1+zL\leq y\leq 1-zL\\ F_{3}\left(y\right) & 1-zL<y\leq zL+1\\ 1 & y>1+zL \end{array}\right.$$
where
(11)$$F_{1}\left(y\right)=\frac{\pi z_{+}+2\sqrt{1-z_{+}^{2}}+2z_{+}\sin^{-1}z_{+}}{4zL\pi },$$
(12)$$F_{2}\left(y\right)=\frac{1}{2}+\frac{\sqrt{1-z_{+}^{2}}-\sqrt{1-z_{-}^{2}}+z_{+}\sin^{-1}z_{+}-z_{-}\sin^{-1}z_{-}}{2zL\pi },$$
(13)$$\begin{array}{c} F_{3}\left(y\right)=\frac{\frac{3\pi zL+\pi y}{2}-z_{-}sin^{-1}\left(z_{-}\right)-\sqrt{1-z_{-}^{2}}}{2zL\pi }, \end{array}$$

*Case 2* (zL=1):(14)$$F\left(y\right)=\left\{\begin{array}{cc} 0 & y<-2\\ F_{1}\left(y\right) & -2\leq y<0\\ F_{2}\left(y\right) & y=0\\ F_{3}\left(y\right) & 0<y\leq 2\\ 1 & y>2 \end{array}\right.$$
where
(15)$$F_{1}\left(y\right)=\frac{\pi \left(0.5+y\right)+2\sqrt{1-{\left(0.5+x\right)}^{2}}+\left(1+2y\right)\sin^{-1}\left(0.5+y\right)}{2\pi },$$
(16)$$F_{2}\left(y\right)=\frac{1}{2},$$
(17)$$F_{3}\left(y\right)=\frac{\frac{3\pi +\pi y}{2}+\left(y-1\right)\sin^{-1}\left(1-y\right)-\sqrt{1-{\left(1-y\right)}^{2}}}{2\pi },$$
and $z_{\pm }=zL\pm y$. The PDF and CDF of *U* and *Y* are shown in Figure 4a,b, respectively. As can be seen in Figure 4a, the range of fabric positions is larger than that of the rigid body, meaning they occupy different probability spaces.

From (9), it is apparent that the CDF of *U* is *independent of ω*, meaning that if the AR task is to classify movements of different frequencies (e.g., $\omega_{1}$ and $\omega_{2}$), *the data distribution cannot increase statistical distance to discriminate between classes*, i.e., $D(U_{\omega_{1}},U_{\omega_{2}})=0$. However, the dependency of the CDF of *Y* on ω is apparent in (10) and (14), leading to a statistical distance between movements of different frequency (to simplify the computation, $\omega_{1}$ and $z_{1}$ represent low frequency, and $\omega_{2}$ and $z_{2}$ represent high frequency). It is apparent that the largest statistical distance is when $y=1+z_{1}L$. The statistical distance is shown below:(18)$$D\left(Y_{\omega_{1}},Y_{\omega_{2}}\right)=\frac{\pi \left(z_{21}L-1\right)+2\sqrt{1-{\left(z_{21}L-1\right)}^{2}}-2\left(1-z_{21}L\right)sin^{-1}\left(z_{21}L-1\right)}{4z_{2}L\pi },$$
where $z_{21}=z_{2}-z_{1}$.

As $D(Y_{\omega_{1}},Y_{\omega_{2}})>0$, the condition (2) is met, suggesting that *AR based on the motion of the fabric* will lead to *higher classification performance*. Moreover, *$D(Y_{\omega_{1}},Y_{\omega_{2}})$ increases monotonically with L*, as can be seen by examining its derivative
(19)$$\begin{array}{c} \frac{dD(Y_{\omega_{1}},Y_{\omega_{2}})}{dL}=\frac{\pi -2(\sqrt{-Lz_{21}\left(Lz_{21}-2\right)}+\sin^{-1}\left(1-Lz_{21}\right))}{4\pi z_{2}L^{2}}, \end{array}$$
where $z_{21}=z_{2}-z_{1}$. Noting that $0<z_{21}L\leq 1$, as $\max\{2\left(\sqrt{-Lz_{21}\left(Lz_{21}-2\right)}+\sin^{-1}\left(1-Lz_{21}\right)\right)\}=\pi$, it is clear that $\frac{dD(Y_{\omega_{1}},Y_{\omega_{2}})}{dL}>0$ for $0<z_{1}L<z_{2}L\leq 1$. This suggests that *the looser the fabric, the greater the statistical distance*. To the authors’ knowledge, this is the first analytical model to capture and explain the empirical finding that data from loose clothing can lead to enhanced AR. An empirical study of this example is provided in the next section.

## 4. Activity Recogntion Via Statistical Methods

As noted in Section 3.2, the extent to which fabric motion helps AR in practice will depend on both the *complexity of the movement* and the *physical properties of the fabric*. In this section, three empirical case studies are presented to test the model’s predictions when using a well-established statistical machine learning approach for AR with fabric-induced motion data. The cases considered are (i) a numerical simulation of the example described in Section 3.3, (ii) its physical realization and (iii) a real human AR task (data and source code for these experiments are available online at https://doi.org/10.18742/22182358 (accessed on 9 May 2023).).

### 4.1. Case Study 1: Simple Harmonic Motion

This evaluation aims to verify the predictions of the proposed model using a numerical simulation.

#### 4.1.1. Materials and Methods

Data were collected from a numerical simulation of the system shown in Figure 2a implemented in MATLAB R2019b (MathWorks, Natick, Massachusetts, USA) consisting of trajectories of length T=2π s generated at a sampling rate of $f_{s}=10$ Hz using (1), from random initial angles ψ∼(−π,π) with *a* = 1 m. Each data set contains *N* = 400 trajectories with 200 trajectories from the yoke running at low frequency (i.e., $\omega_{1}=1\;\mathrm{rad}\;s^{-1}$) and 200 at high frequency ($\omega_{2}=2\;\mathrm{rad}\;s^{-1}$). The same procedure was used to collect fabric movement data Y for lengths $L\in \{\frac{1}{3},\frac{2}{3},1\}m$ (i.e., $F_{1}$, $F_{2}$, $F_{3}$) and rigid body data U (i.e., $R_{1}$). The data were split into equal-sized training and test sets and used to train an SVM classifier (Libsvm toolbox [23]) to perform AR with Gaussian radial basis functions (RBFs) as the kernel function. The SVM was trained to predict the mapping
(20)$$\phi_{n}\mapsto c_{n}$$
in an online fashion, where $\phi_{n}$ is a fragment of the *n*th trajectory and $c_{n}\in \left\{0,1\right\}$ is the corresponding class label (c=0 for $\omega_{1}$, c=1 for $\omega_{2}$). Specifically, following [9], each trajectory is segmented into overlapping windows of size i−1 (where i<K and $K=T/f_{s}$), i.e.,
(21)$$\Phi :=\left(\phi_{1},\phi_{2},\ldots \phi_{N}\right)=\left({\left(y_{1},\ldots ,y_{i}\right)}^{Τ},{\left(y_{2},\ldots ,y_{i+1}\right)}^{Τ},{\left(y_{3},\ldots ,y_{i+2}\right)}^{Τ},\ldots \right).$$The procedure was repeated 100 times for each condition and the classification accuracy computed.

#### 4.1.2. Results

Figure 5a shows the overall accuracy of AR using the SVM classifier with different window sizes. As can be seen, the increased accuracy between the rigid-attached sensor and the fabric-attached sensors is higher when the window sizes are small. Moreover, the accuracy of the four sensors is higher when *L* is greater, in line with the prediction of the model. As the window size increases (i.e., the classifier is given more of the trajectory history), the overall accuracy increases up to 100%, and the increased accuracy between the rigid-attached sensors and the fabric-attached sensors gradually disappears.

### 4.2. Case Study 2: Scotch Yoke

This evaluation aims to validate the proposed framework in a *physical system*.

#### 4.2.1. Materials and Methods

To ensure accurate and repeatable data collection, the experiment reported here uses a physical realization of the system shown in Figure 2a (i.e., an actuated, instrumented *scotch yoke*) as a data acquisition device. The experimental setup is shown in Figure 3c. The mechanism consists of a sliding yoke with rigid rods affixed to either side and a rotating disk with a diameter of 20cm mounted on two bearing blocks driven by a DC motor with encoder (30:1, 37D gear-motor, Pololu Corporation, North Las Vegas, NV, USA) at the fulcrum. The motion of the disk and yoke are coupled via a sliding pin, ensuring a pure sinusoidal movement of the yoke. Affixed to the latter, 10cm away from the fulcrum, was a 30 cm × 5 cm strip of woven cotton fabric, upon which were mounted three sensors (NDI Aurora Magnetic Tracking device, NDI, Canada) that synchronously record the horizontal position at 40Hz at an accuracy of approximately 0.09mm, which were attached along the length of the fabric at (i) 20cm ($F_{2}$), (ii) 30cm ($F_{3}$) and (iii) 40cm from the fulcrum ($F_{4}$) (i.e., at the tip of the fabric). A further sensor ($R_{1}$) was rigidly attached to the yoke at the fabric attachment point (see Figure 3c). The error in the yoke movement against the sinusiodal reference is $0.05\pi \mathrm{rad}\;s^{-1}$ (Details of the error estimation process are given in Section A.2).

With this set up, data were collected from the device driven at the desired speed for the experimental condition (see below) (note that at high speeds, the NDI device occasionally loses track of its sensors, resulting in missing data (i.e., gaps) within trajectories. These are filled using piecewise cubic spline interpolation). Specifically, the following reports the effect of varying (i) the window size $0.025\leq i/f_{s}\leq 2.5\;\;s$, where $\omega_{1}=1.05\pi \;\mathrm{rad}\;s^{-1}$ and $\omega_{2}=1.48\pi \;\mathrm{rad}\;s^{-1}$, and (ii) the difference in frequencies (i.e., $|\omega_{2}-\omega_{1}|$), where $i/f_{s}=0.025\;\;s$ and $\omega_{1}=1.05\pi \;\mathrm{rad}\;s^{-1}$. For this, 30 sample trajectories of length T=5s at each speed were recorded. These data were segmented for online learning through a similar procedure, as described in Section 4.2, randomly split into equal-sized training and test sets and used to train an SVM classifier to perform AR. All data were standardized using the *z*-score. The performance of the classifier was assessed by computing its accuracy and the KS test statistic via (3). This process was repeated for 100 trials of every experimental condition tested.

#### 4.2.2. Results

Figure 5b shows the accuracy of AR using the SVM classifier for different window sizes. As can be seen, a similar but more pronounced trend is seen here, as predicted by the simulation in Section 4.2 (*q.v.*): The accuracy is higher for fabric-attached sensors at small window sizes (and higher for sensors at greater *L*). As the window size increases, the accuracy converges toward the same value, regardless of the sensor used.

Figure 5c,d shows the classification accuracy and KS test statistic D for discriminating between $\omega_{1}=1.05\pi \;\mathrm{rad}\;s^{-1}$ and the different $\omega_{2}$ when $i/f_{s}=0.025\;s$. As can be seen, for the fabric-mounted sensors, the statistical distance *D* is larger. Moreover, the larger the difference between movement frequencies, the greater the statistical distance and therefore the better the performance of AR (Pearson’s correlation coefficient (PCC) between *D* and accuracy is 0.85, indicating a strong positive relationship). For the rigidly attached sensor, there is no obvious increase in *D* or AR performance. This is consistent with the prediction of (2).

### 4.3. Case Study 3: Human Activity Recognition

In this section, the predictions of the proposed framework are evaluated in a human motion recognition task. The AR task chosen for this experiment is that of the recognition of constrained periodic movements (such as operating a crank, winch, or ratchet system) from loose, sensorized clothing (the experiments reported here were conducted with the ethical approval of King’s College London, UK: MRPP-21/22-33739).

#### 4.3.1. Hypothesis

The null hypothesis in this experiment is the classification accuracy of the wrist-attached and sleeve-attached sensors has no significant difference. The alternative hypothesis is the classification accuracy of the wrist-attached sensors and sleeve-attached sensors has significant difference.

#### 4.3.2. Experimental Procedure

The experimental set up (see Figure 6) consists of a *hand winch* with a 15 cm rotating crank handle that experimental subjects must operate at prespecified speeds while wearing a sensorized shirt.

Specifically, participants were asked to remove or roll up the sleeve of any clothing on their arms and to wear an instrumented shirt (96% woven cotton, 4% spandex) with one Aurora sensor attached to the cuff of the right-side sleeve (with a maximum possible displacement from the wrist of ±10cm). They were then directed to operate the winch at *low* ($\omega_{1}=1.25\pi \mathrm{rad}\;s^{-1}$) and *high* frequency ($\omega_{2}=2.5\pi \mathrm{rad}\;s^{-1}$) while their movement was recorded.

All data collection was conducted in an isolated room with no visual or audible distractions. To control the speed, a display screen was used to show the target and actual position of the hand winch in real time (the latency in the display is 0.0078s) as red and blue points, respectively. To accustomize the participant to the set up, the experimenter demonstrated the desired movement in 1 trial, and subjects were directed to perform the target movement for 10 trials at each frequency prior to the experiment. Before data collection, all participants’ sitting positions were standardized. The chair’s height was adjusted to ensure the wrist/hand extension angle was zero when the crank handle moved parallel to the ground. Sleeves were adjusted to ensure that the cuff was aligned flush with the wrist, using clips/tape where necessary.

After the above-mentioned preparatory work was completed, an equal number of sample trajectories (thirty) of length T=8s at high and low frequency were recorded from each participant. During data collection, participants were asked to keep their palms parallel to the ground and to rotate the hand winch to follow the target as closely as possible. Participants were given a 20s rest between each recording. The order of collecting high and low frequencies was alternated between participants (i.e., the first participant performed low-frequency trials first, followed by high-frequency ones; the second participant performed the high-frequency trials first, and so on).

The first 3s of each sample trajectory was discarded to allow for the time it took the participant to adjust to follow the target accurately. The remaining data were used for AR using the method described in Section 4.2 with a window size of 0.2s.

#### 4.3.3. Participants

In the results reported below, 13 healthy volunteers recruited from the local community (6 males, 7 females; mean ± standard derivation; age years old; height 168 ± 11 cm; mass; arm length; wrist circumference 15.3 ± 0.1 cm; laterality index assessed by Edinburgh handedness questionnaire [24]) took part in this experiment. An information sheet was given to all participants before visiting the laboratory, and a consent form was signed by all participants before data collection. To ensure a good, comfortable fit of the instrumented shirt, all participants had an arm length not longer than 78 cm. As the shirt was sensorized on the right-side sleeve, only right-handed participants were recruited.

#### 4.3.4. Results

The relative AR accuracy when using the sleeve-attached sensors as compared with the wrist-attached ones ranged from −0.64% to 8.1% (median 2.1%). Greater accuracy when using the sleeve-attached sensor was seen for 11/13 participants. Using a Wilcoxon matched-pairs signed-rank test, it was seen that there is a statistically significant difference in the accuracy of AR between using wrist- and sleeve-attached sensors (p<0.05). Therefore, the null hypothesis is rejected. The performance of AR between sleeve-attached and wrist-attached sensors had a significant difference. These results indicate a modest average increase in accuracy in this experiment, as might be expected given the relatively small *L* in this task (see Section 3).

## 5. Discussion

This work presents a framework with which to understand the effect of textile motion on AR, including how it may enhance performance compared with the use of rigidly attached sensors. By taking a statistical modeling approach, it is seen how stochasticity in the fabric motion can *amplify* the statistical distance between movement signals, *enhancing AR performance*. The predictions of this model were verified through numerical and physical evaluations, including human motion capture for simple oscilliatory movements, with the findings that it (i) improves as the fabric becomes looser (*L* increases) and that (ii) the discrepancy with rigid sensor use is the most pronounced at small window sizes. The latter suggests that use of fabric-mounted sensors may enable faster and more accurate predictions in the context of *online* AR. The probabilistic model introduced in this paper is the first to predict this effect, and the strong positive correlation relationship between *D* and accuracy in the numerical analysis and empirical findings (Section 4.1) indicate the proposed probabilistic model is a good starting point to predict AR performance in real-world applications. More broadly, the fact that these effects can be analytically modeled with a relatively simple model opens up the possibility of enhanced design and analysis of motion capture systems that use ordinary garments, thereby enjoying a high level of comfort and user-acceptance.

In experiments involving real human movements (see Section 4.3), it is seen that AR based on data from sleeve-attached sensors is more accurate for the majority of participants, albeit the average increase is modest and there is some variability. Possible reasons for this includes the relatively small *L* in relation to the amplitude of the movement. It is notable that the increase in AR accuracy for sensor $F_{2}$ (similarly located 10cm from sensor $R_{1}$) in the experiment reported in Section 4.2 is also modest, in line with the numerical predictions from the proposed model, as predicted by the proposed model. It is also possible that several other factors may play a role, such as the mass of the sleeve material (relatively heavy due to its multilayer design) or the complexity of its geometry (leading to complex internal forces and motion constraints) as compared with the lightweight, simple strip of fabric studied in the scotch yoke experiments. Future work will examine how such factors may be incorporated into the proposed modeling approach to verify whether its predictions can be improved further. Other areas of future work include extending the model to account for external factors that might affect fabric movement, such as wind, humidity, impact of the fabric with objects in the environment or different fabric materials (e.g., polyester). Moreover, the analytical modeling can also be extended to multidimensional movements, such as may be required for full-body motion analysis. These, in turn, may open up many applications in robotics and automation, such as analyzing workers’ behavior in manufacturing lines [25,26] and human–robot interaction (e.g., for control of exoskeletons [27] or prostheses [28], etc.).

## Figures and Tables

**Figure 1 sensors-23-04669-f001:**
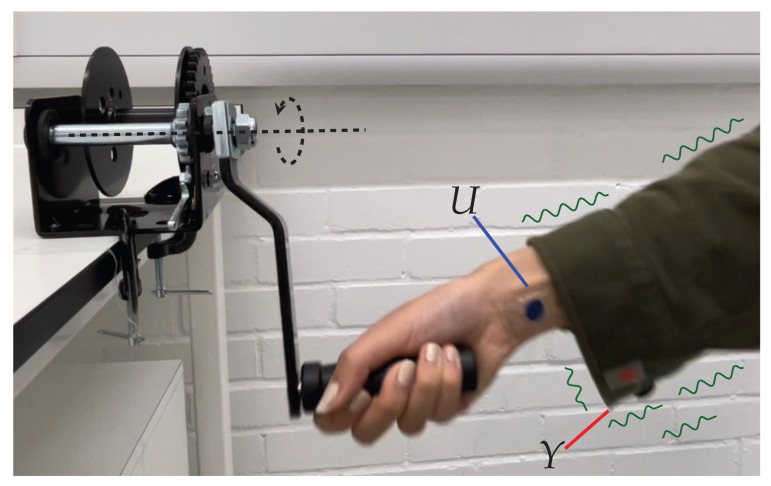
New sensing technologies have led to the potential to capture human movement from clothing-embedded sensors *Y* instead of relying on those rigidly attached to the body *U*. However, use of ordinary garments exposes the former to additional and unpredictable artifacts in the signal.

**Figure 2 sensors-23-04669-f002:**
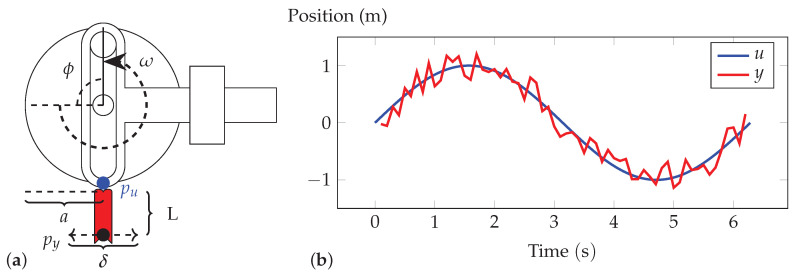
(**a**) A scotch yoke mechanism with a piece of fabric (red) attached. Sensors are affixed at $p_{u}$ (the tip of the sliding yoke) and $p_{y}$ (the tip of the fabric). (**b**) Simulated signals of the sensors from this model (zL=0.3).

**Figure 3 sensors-23-04669-f003:**
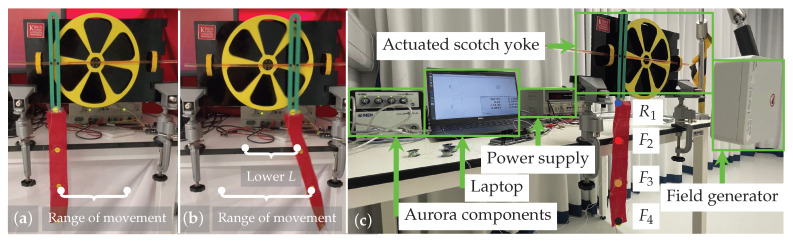
The scotch yoke mechanism with a piece of fabric attached moving at (**a**) low and (**b**) high frequency. (**c**) Front view of the experiment set up. Sensors are placed (i) at equidistant intervals along the fabric strip ($F_{2}$–$F_{4}$) and (ii) rigidly at the attachment point ($R_{1}$).

**Figure 4 sensors-23-04669-f004:**
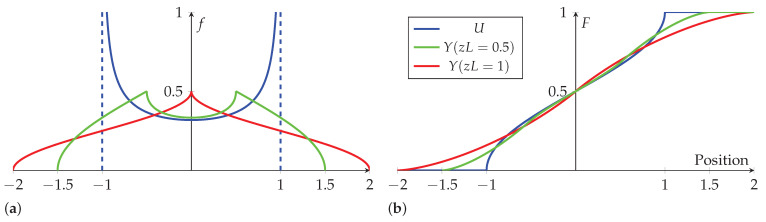
The (**a**) PDF and (**b**) CDF of rigid body and fabric position (zL=0.5,1).

**Figure 5 sensors-23-04669-f005:**
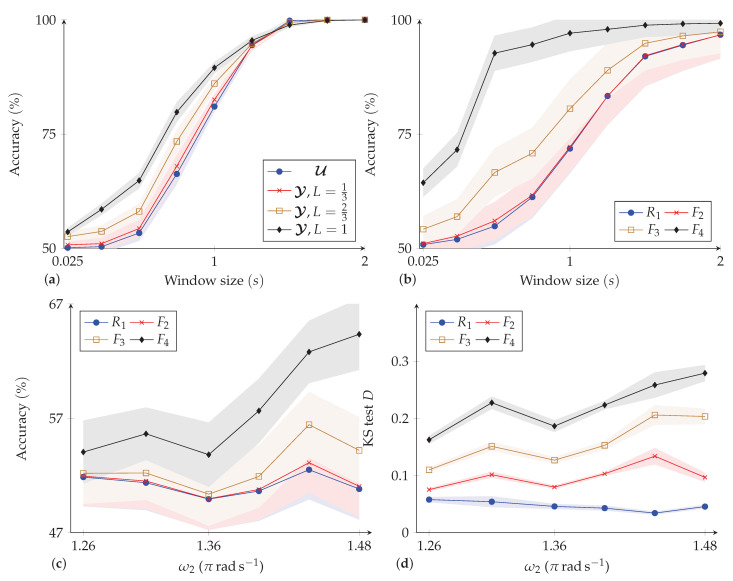
Accuracy in recognizing movements of different frequencies in the (**a**) simulated and (**b**) physical scotch yoke when varying window size; accuracy (**c**,**d**) test statistic *D* for different pairs of frequencies with window size 0.025s. Reported are the mean ± s.d. (the shaded area) over 100 trials.

**Figure 6 sensors-23-04669-f006:**
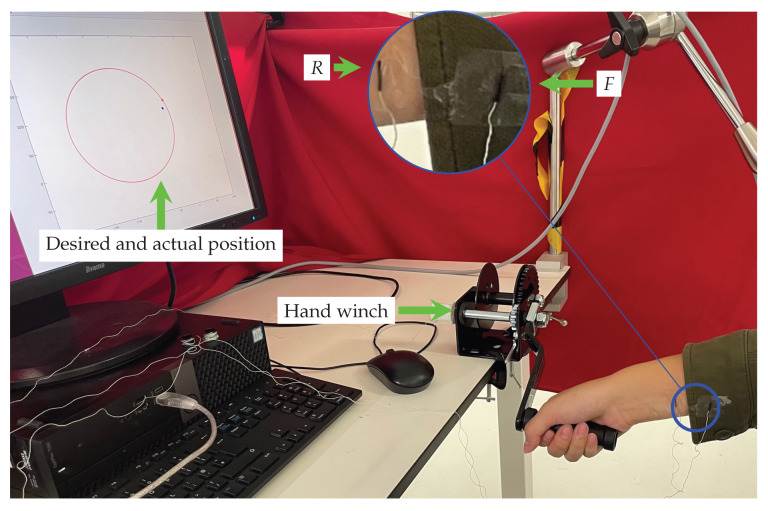
Experimental set up for the crank task. The participant operates a hand winch at prespecified speeds while their motion is captured with a wrist-attached *R* and garment-mounted sensor *F*.

**Table 1 sensors-23-04669-t001:** Selected state-of-the-art sensing on clothes research.

Reference	Sensor Type	Sensor Placement	Type of Clothes or Fabric	Activities	Method	Strength/Finding	Limitation
Jayasinghe et al. [10]	accelerometers	waist, thigh, ankle and clothes in similar position	slacks, skirt and frock	four daily activities	correlation coefficients decision tree	clothing and body worn sensor data are correlated	sensors are heavy
Jayasinghe et al. [11]	IMU	waist, thigh, lower shank and clothes in similar position	daily clothes	gait cycle	correlation coefficients	clothing worn sensor data have key points in the gait cycle	classification accuracy of the sensors has not been investigated
Jayasinghe et al. [17]	IMU	waist, thigh, ankle and clothes in similar position	daily clothes	4 static and 2 dynamic activities	KNN	clothing worn sensor has good posture classification	the number of participants is limited
Michael et al. [9]	accelerometers	rigid pendulum and a piece of fabric attached	denim, jersey and roma	low and high swing speed	SVM, DRM	fabric sensors has higher accuracy of AR	lack theoretical model
Bello et al. [13]	capacitive	four antennas to cover the chest, shoulders, back and arms	loose blazer	20 posture/gestures	conv2D	sensor is not affected from muscular strength	affected from conductors
Cha et al. [14]	piezoelectric	clothing near knee, hip	loose trousers	gait	rule-based algorithm	feasibility of gait detection	gender of participants is not balanced
Skach et al. [15]	pressure	clothing near thigh	loose trousers	19 postures	random forest	sensor can detect human postures	the upper body has not been tested
Lin et al. [16]	strain	clothing near shoulder, elbow, waist and abdomen	loose jacket	daily activities, postures and slouch	CNN-LSTM	sensor can detect human postures	gender of participants is not balanced
Tang et al. [18]	strain	several positions on the body	juncus effusus fiber	several daily activities	gauge factor	sensitive stretchable	the maximum sensing range is limited
Lu et al. [19]	strain	human joints	conductive PSKF@rGO	exercise monitoring	gauge factor	useful under extreme conditions	the strain range is limited
Xu et al. [20]	strain	various human body parts	composite fiber	language recognition pulse diagnosis	gauge factor	sensitivity, stability and durability	the number of participants is limited
**Our approach**	magnetic	scotch yoke with a piece of fabric attached wrist and sleeve	woven cotton	various rotating frequencies	SVM		

## Data Availability

All data and source code in this paper is available online at https://doi.org/10.18742/22182358.

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
