# Peer review of "A Probabilistic Model of Human Activity Recognition with Loose Clothing†"

_sensors, 2023, doi:10.3390/s23104669_

Round 1

Reviewer 1 Report

The study proposed a probabilistic model that explains improved responsiveness and accuracy with fabric sensing from the increased statistical distance between movements recorded. The work aimed for limited scope of work in objectives. So the experiments are achieved. Introduction section needs more background study to present. The significance of the work needs to be related to problems in the state of art. 

Author Response

Many thanks for your comments. We have refined the introduction (see lines27-33). We also explain how the significance of the work is related to problems in the state of the art (see lines27-33).

Reviewer 2 Report

This work presents a framework with which to understand the effect of textile motion. A probabilistic model was proposed and verified. The stochasticity in the fabric motion were studied to show how it can amplify the statistical distance between movement signals. AR performance is also enhanced. The study is interesting and systematic. The writing is good. I therefore recommend publication of this paper. 

Author Response

Many thanks for your comments.

Reviewer 3 Report

The authors present an interesting paper where well describes a probabilistic model to recognize wrist activity.

The work is overall complete and well structured. The session in which the preliminary device calibration activities were reported was excellent.

Author Response

Many thanks for your comments.

Reviewer 4 Report

This manuscript proposed a probabilistic model that explained improved responsiveness and

accuracy with fabric sensing compared to use of rigidly-attached sensors. This work demonstrated comprehensive details combined with theoretical explanation and experimental invalidation. Overall, this work is interesting and done well. However, some major points need to be clarified or revised. Below its my comments for consideration.

1.     The background demonstrated in the abstract could be refined and some contents of this research should be added as well to highlight the novelty of this work.

2.     There are some publications about fiber-based smart sensors to sense strain signal, and the author may cite them in the manuscript to enrich the introduction. For example: Chemical Engineering Journal, 2022, 438, 135600; Adv. Fiber Mater., 2023, 5, 223; Adv. Fiber Mater., 2023, 5, 282.

3.     The keywords should be revised and supplemented, which may not fully cover this research.

4.     Strengths and weaknesses of related work could be provided in the Table 1.

5.     Most of previous research placed the sensor on clothing near to joints. What is the advantage of the designed sensor placement method “scotch yoke with scotch yoke with

a piece of fabric attached”?

6.     The revealed experiments and models were based on specific movements which induced  monotonous sensing signals, however human activities are always complex and random. This should be taken into accounts. In other words, how is the practicability of this model?

7.     Authors have considered that the physical properties of the involved fabric will have influences on the AR in practice, which is a good point. Thus, it may be instrumental to add some comparison to further verify the reliability of this model. For instance, polyester which has been recognized as the second-widely-used fabrics could be employed for comparison.

8.     Any insights into the mechanism that “As the window size increases, the overall accuracy increases and the difference between fabric and rigid data gradually disappears”?

9.     Error bars should be added in Figure 5 since these trails were repeated for several rounds.

10.   How to design a model to decouple and recognize one activity from multiple movements? This should be considered for future work.

Author Response

Many thanks for your comments. Please see the response letter attached. 

Reviewer 5 Report

SENSORS-2222634

Overall Comments

This study proposes a probabilistic model that explains that using sensory data from attached to fabric can help human activity recognition. This study shows that human activity can be better classified by using fabric attaches sensors, allowing monitoring of additional movements than from sensors attached to a rigid body part. However, the reviewer has concerns that need to be addressed before its publication. Overall, the manuscript lack of information, including the problem statement, detailed information about the data collection, the explanation of the results and model performance measurement, and the importance and contribution of this research for AR. Furthermore, This study validates the developed model by classifying different frequencies of movements for the same activity, rather than different activities. From this validation, the conclusion that monitoring/using fabric motion will lead to higher performance of activity recognition tends to be overgeneralized.

  1. Introduction

    1. For line 25, it would be better to provide an intuitive explanation of how the fabric motion may assist human motion analysis.

    2. For lines 77-82, how this result demonstrates that fabric movement may be beneficial to human activity recognition?

  1. Probabilistic Modelling Framework

    1. For lines 175-187 and for lines 184-185, in many cases, for AR, bodily movements are monitored by accelerometer and gyroscope sensors, which can detect different movements with the same frequency of movement.

    2. While distance and frequency information along with bodily movements is also important in activity recognition, it should not be generalized that the fabric movement leads to higher AR performance because it excludes speed and rotational body part movements.

    3. What types of the sensor was used for this study?

  1. Probabilistic Modelling Framework

    1. For lines 222-224, it read confused. Is the accuracy higher with a small window size? or a long window size?

    2. For Case Study 3, are the labels low, medium, and high frequencies? This experiment is to differentiate different frequencies of movements, rather than activity recognition.

    3. Can this approach be exploited for daily activity recognition?

Author Response

(The authors gave the same response as above.)

Round 2

Reviewer 4 Report

The authors have revised the manuscript carefully based on referees' comments. The scientific quality of this paper is greatly improved. The manuscript can be accepted at the present version. 

Author Response

Many thanks for your help and positive response.

Reviewer 5 Report

This study does not seem to have a clear contribution to human activity recognition. While the developed model has been validated to distinguish the same movement with different frequencies, it is difficult to argue that these results would help to recognize daily activities, which involve a series of various movements in sequential order. While distinguishing between motions of different frequencies can help to distinguish some activities, such as walking and running, this study does not show that the developed model can help distinguish such activities rather than simple movement. Therefore, the reviewer believes that the study lacks a sufficient contribution to fill the knowledge gap in human activity recognition.

Author Response

Many thanks for your comments. Please read the attached document.
